# Machine learning-based tissue of origin classification for cancer of unknown primary diagnostics using genome-wide mutation features

Luan Nguyen [1], Arne Van Hoeck [1] & Edwin Cuppen [1,2 ✉]

Cancers of unknown primary (CUP) origin account for ~3% of all cancer diagnoses, whereby the tumor tissue of origin (TOO) cannot be determined. Using a uniformly processed dataset encompassing 6756 whole-genome sequenced primary and metastatic tumors, we develop Cancer of Unknown Primary Location Resolver (CUPLR), a random forest TOO classifier that employs 511 features based on simple and complex somatic driver and passenger mutations. CUPLR distinguishes 35 cancer (sub)types with ~90% recall and ~90% precision based on cross-validation and test set predictions. We find that structural variant derived features increase the performance and utility for classifying specific cancer types. With CUPLR, we could determine the TOO for 82/141 (58%) of CUP patients. Although CUPLR is based on machine learning, it provides a human interpretable graphical report with detailed feature explanations. The comprehensive output of CUPLR complements existing histopathological procedures and can enable improved diagnostics for CUP patients.

[1] University Medical Center Utrecht, Universiteitsweg 100, 3584 CG Utrecht, The Netherlands. [2] Hartwig Medical Foundation, Science Park 408, 1098 XH Amsterdam, The Netherlands. ✉email: e.cuppen@hartwigmedicalfoundation.nl

Cancers of unknown primary (CUPs) is an umbrella term for advanced-stage metastatic tumors for which the tumor tissue of origin (TOO) cannot be conclusively determined based on routine diagnostics (typically via histopathology[1]), and there is also a significant fraction of patients with indeterminate or differential diagnoses, especially with poorly differentiated tumors[2]. Patients with uncertain TOO diagnoses suffer from a lack of therapeutic options as primary cancer type classification is a dominant factor in guiding treatment decisions[3].

Thus far, TOO classifiers have been developed on data from a wide range of molecular methods including DNA sequencing (targeted[4], whole exome[5], and whole genome[6,7]), RNA profiling (from coding RNA[8], microRNA[9–11], as well as whole transcriptome profiling[12,13]), and methylation profiling[14]. Driven by its ability to comprehensively capture actionable biomarkers that enable precision medicine[15], whole-genome sequencing (WGS) is maturing rapidly as a diagnostic tool[16] and is increasingly adopted in clinical systems in various countries[17–19], and could thus be an interesting basis for diagnostic TOO classifiers. Recently developed WGS-based classifiers[6,7] were shown to outperform targeted or whole-exome sequencing-based approaches[4,5] due to being able to utilize mutations from all genomic regions. The main features employed by these classifiers included mutational signatures which are patterns of somatic mutations resulting from exogenous or endogenous mutational processes (e.g., C > T mutations due to ultraviolet radiation exposure in melanoma)[20], as well as regional mutational density (RMD) which represents the genomic distribution of somatic mutations that are associated with tissue type-specific chromatin states, whereby late-replicating closed chromatin regions show increased mutation rates[21]. However, not all WGS-based features are yet fully explored for TOO classification including complex mutagenic features such as viral DNA integrations, driver gene fusions, and other complex structural events (e.g., chromothripsis), as well as non-mutagenic features such as gender, all of which have been shown to be correlated with specific tumor type(s). Indeed, human papillomavirus (HPV) sequence insertions are specifically and frequently found in cervical and head and neck cancer[14], *KIAA1549-BRAF* fusions in pilocytic astrocytomas[13], and liposarcomas frequently harbor *FUS-DDIT3* fusions[15] as well as chromothripsis events[22].

Here we describe the development of CUPLR (Cancer of Unknown Primary Location Resolver), a TOO classifier that integrates current state-of-the-art WGS-based mutation features, including complex structural variant (SV) features. CUPLR comprises an ensemble of binary random forest classifiers that each discriminate one of 35 cancer types with an overall recall of 90%. We find that while RMD and mutational signatures were highly predictive of cancer type (in line with existing classifiers[6,7]), the incorporation of SV features improves prediction performance for cancer types that currently lack highly informative features. Furthermore, we have ensured that the output of CUPLR, namely the prediction probabilities and the features supporting each prediction, are humanly interpretable to facilitate diagnostic use and clinical decision-making with CUPLR.

## Results

**Extraction of genomic features**. To develop CUPLR, we constructed a harmonized dataset from two large pan-cancer WGS datasets from the Hartwig Medical Foundation (Hartwig) and Pan-Cancer Analysis of Whole Genomes consortium (PCAWG)[23]. The raw sequencing reads were analyzed with the same mutation calling pipeline to construct a catalog of uniformly called simple and complex mutations. The harmonized dataset consisted of tumors from 6756 patients across 35 different cancer types (Fig. 1a, Supplementary data 1). In contrast to many previously published papers[4–7], this dataset includes a large proportion of samples taken from metastatic lesions, which is relevant for TOO classification as CUP samples are by definition from patients with metastatic cancer.

A wide range of features ($n = 4131$) were extracted for classifying cancer types based on driver/passenger and simple/complex mutations (Fig. 1c). First, we determined the presence of gain of function (amplifications and activating mutations) and loss of function (deep deletions and biallelic loss) events in 203 cancer-associated genes. These genes were selected based on having enrichment of gain and/or loss of function events in at least one cancer type (see methods). Second, we calculated the mutational load of single base substitutions (SBS), double base substitutions (DBS), and indels for each sample. Third, we determined the number of contributing mutations to the SBS, DBS, and indel signatures from the COSMIC catalog[20]. Fourth, the number of SBSs in each 1 Mb bin across the genome ($n = 3071$) was calculated to determine the RMD[24]. Mutational signatures and RMD were normalized by the mutational load of the respective mutation type to account for differences in mutational load across samples. Fifth, copy number data was used to infer the genome ploidy, diploid proportion, whole-genome duplication status, and gender for each sample[25]. Sixth, for each sample, we determined the copy number change of each chromosome arm relative to the genome ploidy[26]. Lastly, we parsed the called simple and complex SVs to determine: (i) the total SV load per sample; (ii) the number of deletions, duplications stratified by length; (iii) the number of complex events stratified by size; (iv) the size of the largest complex event, (v) the number of long interspersed nuclear element (LINE) insertions and double minutes; and (vi) the presence of gene fusions and viral sequence insertions[25,27].

**Classifier training**. The extracted genomic features were then used to develop CUPLR, a classifier consisting of two components (Fig. 1d). The first component is an ensemble of binary random forest classifiers each discriminates one cancer type versus other cancer types (i.e., one-vs-rest). We chose to use an ensemble of binary classifiers as opposed to one multiclass classifier so that feature selection could be performed per cancer type, since different features are important for each cancer type. Additionally, we chose to use random forests over other algorithms (e.g., neural networks) as they can natively handle different feature types (continuous, boolean, categorical, etc) without requiring feature values to be scaled, which also improves model interpretability. The second component of CUPLR is an ensemble of isotonic regressions to calibrate the probabilities produced by each random forest. Random forests tend to be overconfident at probabilities towards 0 and underconfident at probabilities towards 1, and this bias varies between random forests[28]. The calibration we have performed here ensures that probabilities are comparable between random forests. Furthermore, calibration allows for the probabilities to have the following intuitive interpretation: a probability of e.g., 0.8 means that there is an 80% chance of a prediction being correct (this relationship does not hold for the raw "probabilities" from random forests).

We used 6082 samples for training and held out 674 samples as an independent test set, with both having the same cancer type and cohort proportions (Supplementary data 2). Training of the main random forest ensemble involved several steps (Supplementary figs. 1 and 2). Briefly, due to the sheer number of and sparsity of the RMD bins (3071), non-negative matrix factorization (NMF) was performed on the RMD bins for each cancer type to reduce the bins to 46 cancer type-specific RMD profiles. Then,

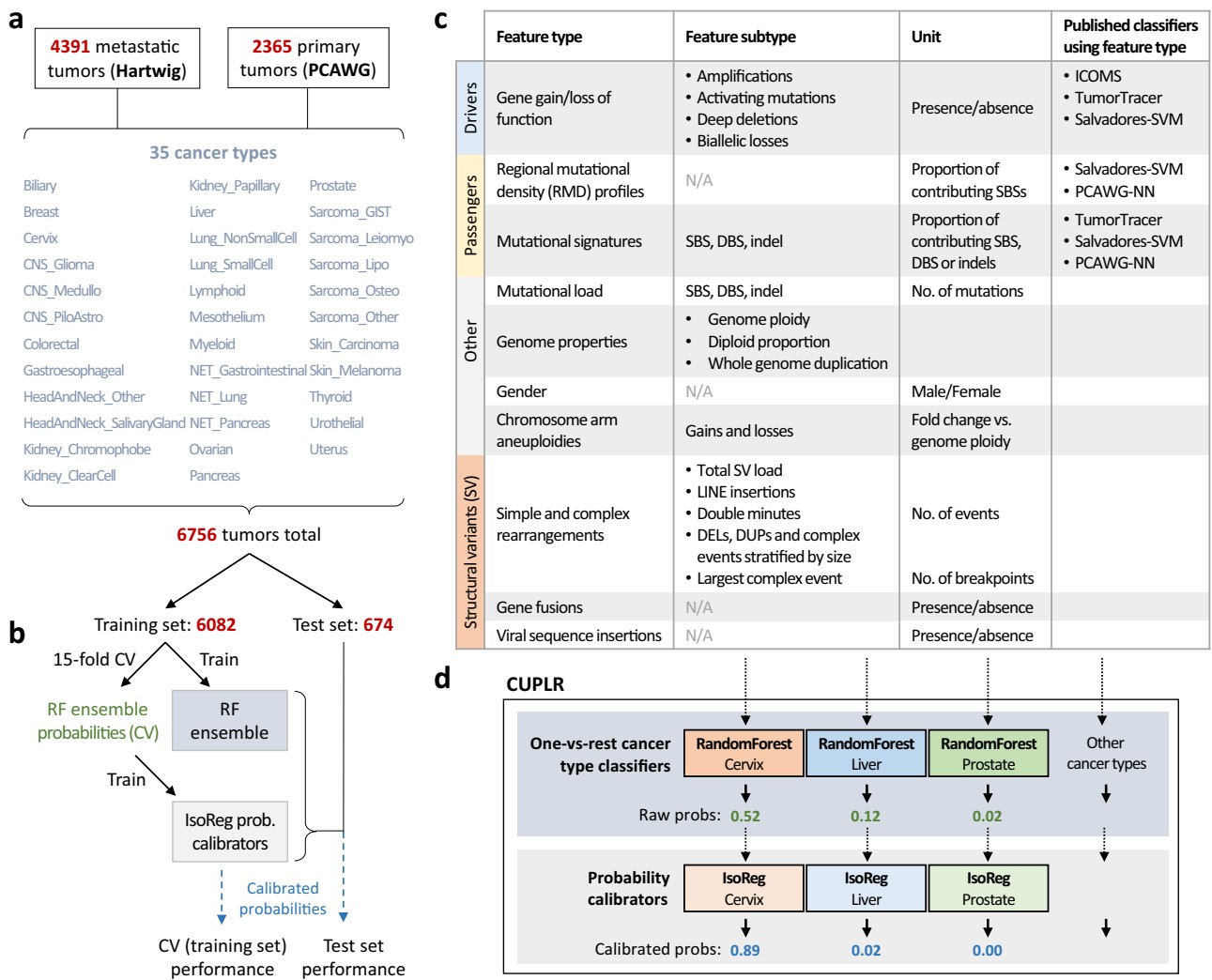

**Fig. 1 Cancer of Unknown Primary Location Resolver (CUPLR) classifies 35 different cancer types using features derived from all mutation types.**
**a** CUPLR was developed using whole-genome sequencing data 4391 metastatic tumors from the Hartwig Medical Foundation (Hartwig) and 2365 primary tumors from the Pan-Cancer Analysis of Whole Genomes (PCAWG) consortium, totaling 6756 samples across 35 different cancer types. **b** 6082 samples were used to train CUPLR and 674 were held out as an independent test set. The whole training set was used to train the final random forest ensemble. 15-fold cross-validation was performed to obtain the random forest cancer type probabilities on the training set, which were then used to train the ensemble of isotonic regressions (for probability calibration). CUPLR is composed of the random forest and isotonic regression ensembles as shown in **d**. The performance of CUPLR was assessed using the calibrated cross-validation probabilities as well as probabilities obtained by applying CUPLR to the test set. **c** A summary of the genomic features extracted from the whole-genome sequencing data and used by CUPLR. Detailed descriptions of each feature can be found in Supplementary data 3. The names of the published classifiers refer to the following studies: ICOMS Inferring Cancer Origins from Mutation Spectra, Dietlen et al.[4], TumorTracer Marquard et al.[5], Salvadores-SVM support vector machine by Salvadores et al.[7], PCAWG-NN PCAWG neural network by Jiao et al.[6]. Cancer type abbreviations: CNS central nervous system, CNS_Medullo medulloblastoma, CNS_PiloAstro pilocytic astrocytoma, NET neuroendocrine tumor, Sarcoma_GIST gastrointestinal stromal tumor, Sarcoma_Leiomyo leiomyosarcoma, Sarcoma_Lipo liposarcoma, Sarcoma_Osteo osteosarcoma, Sarcoma_Other sarcomas other than leiomyosarcoma, liposarcoma or gastrointestinal stromal tumors. Other abbreviations: RF random forest, IsoReg isotonic regression, CV cross-validation, SBS single-base substitutions, DBS double base substitutions, SV structural variants, DEL structural deletions, DUP structural duplications, LINE long interspersed nuclear element.

for each cancer type, univariate feature selection was performed (to remove irrelevant features) with 511 features ultimately being selected (232 numeric and 279 boolean; Supplementary data 3). This was followed by class resampling (to alleviate imbalances in the number of samples for each cancer type), and subsequently training of the binary random forest itself. The above training procedure was applied to all samples of the training set to produce the final random forest ensemble. The random forest ensemble training procedure was then subjected to stratified 15-fold cross-validation to obtain cancer type probabilities for the training samples. These probabilities were then used to train the

ensemble of isotonic regressions for calibrating the random forest probabilities (Fig. 1b, Supplementary fig. 3). Calibration resulted in less biased reliability curves (Supplementary fig. 4) and improved recall, especially for cancer types with few training samples (Supplementary fig. 5).

**Performance of CUPLR.** To assess the performance of CUPLR, we used the cancer type predictions based on the isotonic regression calibrated cross-validation (CV) probabilities, as well as the predictions upon applying CUPLR to the held-out test set

(Fig. 1b). Both the training set ($n = 6082$) and the held-out test set ($n = 674$) had the same cancer type and cohort distribution (Supplementary data 2). CUPLR could predict TOO with 90% (CV) and 89% (test set) overall recall, and 90% (CV) and 89% (test set) overall precision (Fig. 2b, c). Differences between CV and test set recall and precision in certain cancer types were due to low sample sizes in the test set (Fig. 2a, b, Supplementary fig. 6).

High misclassification rates for certain cancer types could likely be explained by shared cancer type characteristics (Fig. 2c). This could be due to a common developmental origin, such as with Uterus being misclassified as Ovarian (CV: 7%, test: 29%) due to both being gynecological cancers[29], and Biliary being misclassified as Pancreas (CV: 24%, test: 42%) and Liver (CV: 9%) due to being cancers of the foregut[30,31]. Cancer subtypes were also often misclassified as other subtypes, which was the case for Lung_SmallCell towards Lung_NonSmallCell (CV: 40%, test: 60%); Kidney_Papillary towards Kidney_ClearCell (CV: 38%, test: 67%); and Sarcoma_Leiomyo (CV: 35%, test: 43%) and Sarcoma_Osteo (CV: 17%) towards Sarcoma_Other (sarcomas other than leiomyo-/lipo-/osteosarcomas or gastrointestinal stromal tumors). Neuroendocrine tumor (NET) subtypes were occasionally misclassified as each other, such as NET_Lung towards NET_Gastrointestinal (CV: 9%) and NET_Pancreas (CV: 9%, test: 33%), and NET_Gastrointestinal towards Pancreas (CV: 6%) which may (at least partially) reflect cancer type misannotation amongst these samples due to neuroendocrine tumors having similar morphological features[32]. Likewise, HeadAndNeck_SalivaryGland samples that were misclassified as breast (CV: 23%, test: 33%) were potentially misannotated due to being adenoid cystic carcinomas (i.e. salivary gland-like cancers) of the breast[33].

Thus far, we have mainly assessed performance based on whether the highest probability cancer type is the correct cancer type (i.e. recall; Fig. 2b, c). However, if we consider whether the correct cancer type is amongst the top-2 highest probabilities (top-2 recall; Fig. 2b), overall recall increases from 90% to 95% (CV) and 89% to 94% (test set), with the greatest increases being for the cancer subtypes including Lung_SmallCell (CV: 50% to 83%, test: 40% to 100%), Kidney_Papillary (CV: 62% to 79%, test: 33% to 100%), Sarcoma_Leiomyo (CV: 56% to 89%, test: 57% to 86%) and Sarcoma_Other (CV: 63% to 89%, test: 54% to 77%). A large gain in recall was also observed for the Biliary (CV: 52% to 73%, test: 42% to 83%) which was often misclassified as Pancreas. Similar increases in the recall were seen based on predictions on the test set. The top-2 (and even top-3) probabilities of CUPLR can be particularly useful for differential diagnosis purposes to narrow down potential TOOs when routine diagnostics are not fully conclusive.

**Added predictive value of SV related features**. When examining the most important feature types from each random forest within CUPLR (Fig. 3a), RMD profiles ('rmd') were consistently the most predictive of cancer type (in line with the findings from Jiao et al. 2020[6]), as well as mutational signatures ('sigs') including those with known cancer type associations such as SBS4 (associated with smoking[20]) in lung cancer (Fig. 3b). As these mutation features are derived from genome-wide SBSs and indels, we assessed whether the presence of certain confounding factors that affect the SBS and indel genomic landscape (including DNA repair deficiencies[34,35], chemotherapy treatment[36,37], smoking history[38]) may lead to more incorrect predictions. However, these confounding factors showed minimal impact on classification performance (Supplementary Note 1, Supplementary table 1, Supplementary fig. 13).

In addition to RMD profiles and mutational signatures, gender (as expected) was highly important for predicting cancers of the reproductive organs including breast, cervical, ovarian, prostate, and uterine cancer (Fig. 3b, Supplementary fig. 7). Notably, SV-related features were important for classifying certain cancer types (Fig. 3b). This included known cancer type-specific gene fusions such as *TMPRSS2-ERG* for Prostate[39], *EML4-ALK* for Lung_NonSmallCell[40], *KIAA1549-BRAF* for CNS_PiloAstro (pilocytic astrocytomas)[41], and *FUS-DDIT3* for Sarcoma_Lipo[42]. We also find known viral DNA integrations as important features, including from human papillomavirus (viral_ins.HPV) in Cervix and HeadAndNeck_Other (non-salivary gland head and neck cancers)[43], Epstein-Barr virus in HeadAndNeck_Other[44], hepatitis B virus (viral_ins.HBV) in Liver[45], and Merkel cell polyomavirus (viral_ins.MCPyV) in Skin_Carcinoma[46]. Lastly, the largest complex SV cluster (i.e., by number of breakpoints) (sv.COMPLEX.largest_cluster) which we use as a proxy for the presence of chromothripsis was also predictive for liposarcomas, which are known to frequently harbor chromothripsis events. However, in contrast to the features mentioned above, the presence of chromothripsis alone is not sufficient for classifying a tumor as liposarcoma as chromothripsis is also highly prevalent in other tumor types[22].

To further assess the added value of SV-related features, we excluded entire feature types from the training and examined the decrease in classifier performance (Supplementary fig. 8). Indeed, removal of the viral integration features ('viral_ins') led to a decreased recall for Skin_Carcinoma (70% to 61%) and Cervix (89% to 83%), likely due to loss of the viral_ins.MCPyV and viral_ins.HPV features respectively. Removal of the simple and complex SV features ('sv') resulted in a drop in recall for Sarcoma_Leiomyo (56% to 49%), likely as the size of the largest complex SV cluster (sv.COMPLEX.largest_cluster) can discriminate Sarcoma_Lipo and Sarcoma_Osteo from Sarcoma_Leiomyo (Supplementary fig. 7). Lastly, removal of the gene fusion features ('fusion') resulted in a large decrease in recall for Lung_SmallCell (50% to 38%) likely as *EML4-ALK* fusions are characteristic of non-small cell (but not small cell) lung cancer.

When compared to existing WGS-based CUP classifiers[5–7], CUPLR is able to classify more cancer (sub)types and showed improved recall and precision (Supplementary figs. 9 and 10) for cancer types where SV-related features were important, including for CNS_PiloAstro, Lung_NonSmallCell, and Prostate. Overall, CUPLR achieved a similar recall to existing classifiers for the remaining cancer types (except for head and neck, myeloid, pancreatic neuroendocrine, and thyroid cancers). Likewise, precision was also similar to other classifiers for the remaining cancer types (except for head and neck, myeloid, thyroid, and uterine cancers).

In summary, the incorporation of all feature types resulted in the best performance, with SV-related features being important for specific cancer types.

**Graphical prediction report**. Aside from cancer type probabilities, CUPLR also outputs explanations as to which features support these probabilities. These allow one to verify the predictions based on existing knowledge, and could be included in diagnostic reports to support decision making, e.g., in molecular tumor boards. The feature explanations are based on feature contribution calculations which enable feature importance determination at the sample level (rather than at the cohort level as in Fig. 3). Specifically, feature contributions represent the total gain (or loss) in probability upon passing a feature through a random forest[47]. To ease the interpretation of CUPLR's output, we have implemented a graphical report (Fig. 4) which can be generated per patient that shows the cancer type probabilities (left panel), feature contributions for the top features for the top

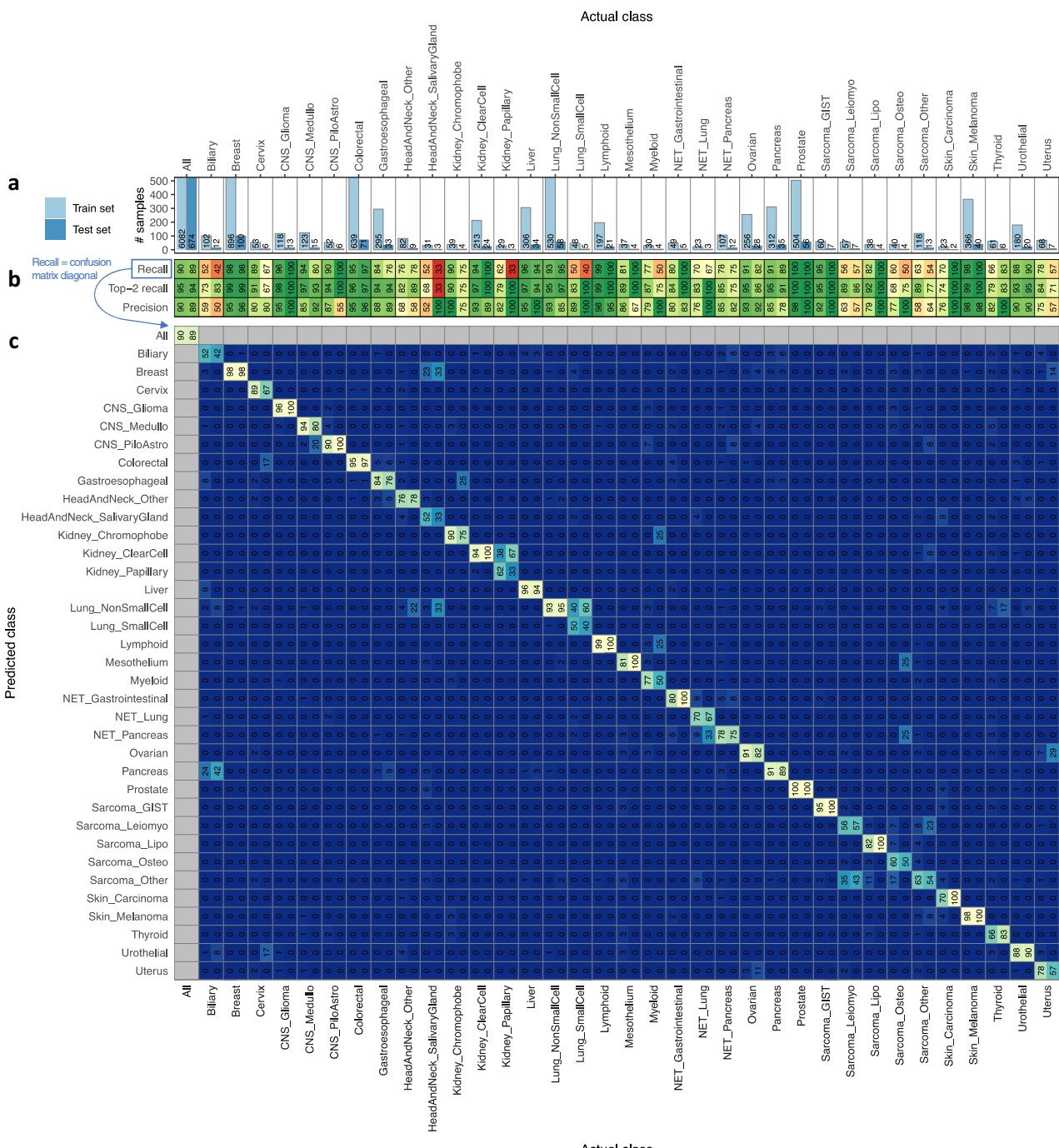

**Fig. 2 Performance of CUPLR. a** The total number of samples for each cancer type for the training set (left columns) and held out test set (right columns). Note that the y axis of **a** is truncated to better visualize cancer types with small sample sizes. **b**, **c** also have the same left/right column layout corresponding to the training set cross-validation and test set performance. **b** Summary of performance metrics. Recall: % of correctly classified samples per cancer type, and is equivalent to the diagonal values in **c**. Top-2 recall: % of correctly classified samples when considering the 2 highest probability cancer types as correct. Precision: % of correctly classified samples amongst samples predicted as a particular cancer type. Overall performance metrics (i.e. under the 'All' column) are micro-averages. **c** Confusion matrix showing the performance of CUPLR where columns represent the % of samples in a cancer type cohort predicted as a particular cancer type. The diagonal represents the % of samples correctly predicted as a particular cancer type (equivalent to recall). Raw data for performance metrics and confusion matrix in **b**, **c** can be found in Supplementary data 4. Cancer type abbreviations: CNS central nervous system, CNS_Medullo medulloblastoma, CNS_PiloAstro pilocytic astrocytoma, NET neuroendocrine tumor, Sarcoma_GIST gastrointestinal stromal tumor, Sarcoma_Leiomyo leiomyosarcoma, Sarcoma_Lipo liposarcoma, Sarcoma_Osteo osteosarcoma, Sarcoma_Other sarcomas other than leiomyosarcoma, liposarcoma or gastrointestinal stromal tumors.

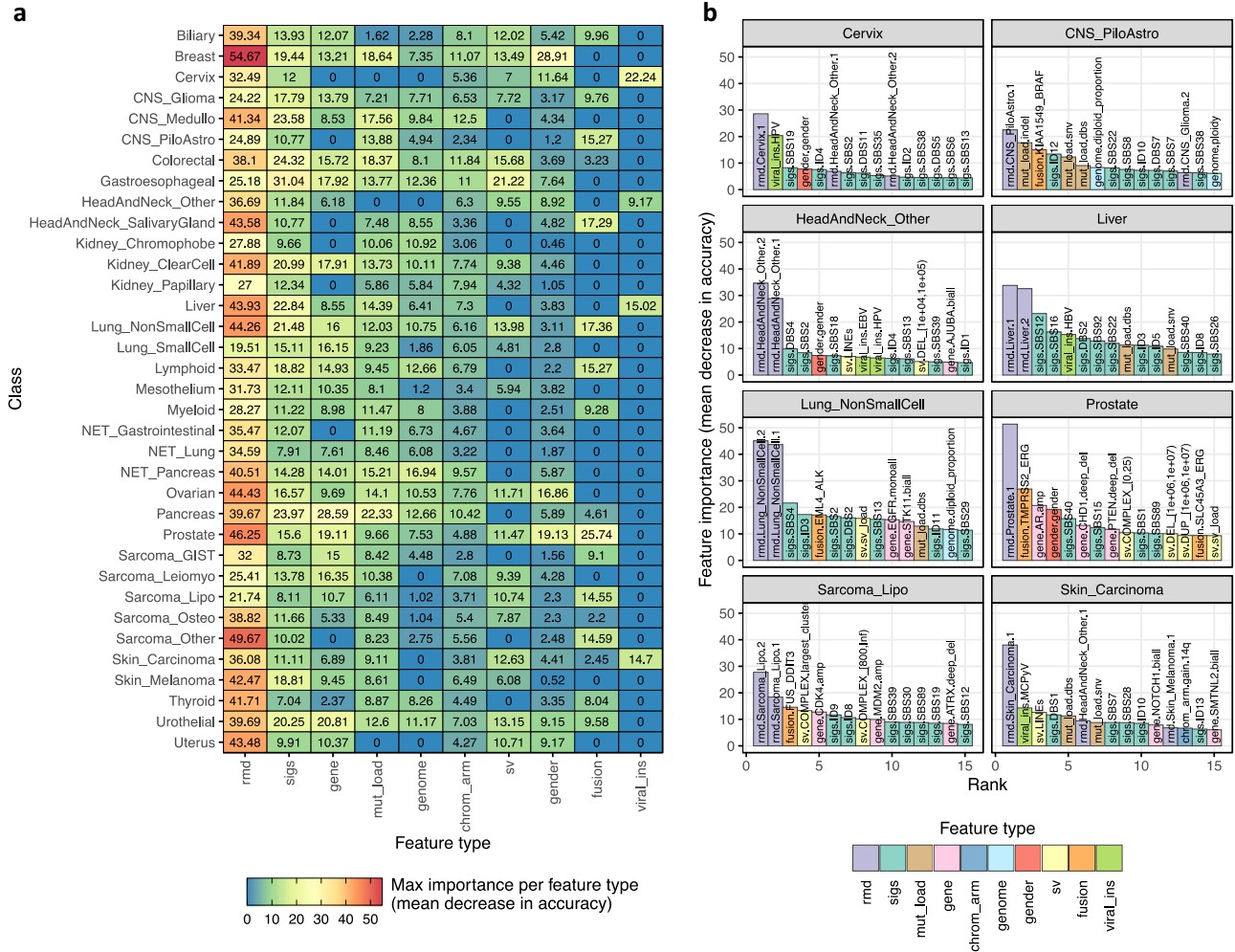

**Fig. 3 Features most predictive of cancer type. a** Maximum feature importance per feature type for each cancer type random forest from CUPLR. See Supplementary data 3 for the descriptions as well as feature importance values for each feature. **b** Feature importances from the top 15 features for selected cancer type random forests. Feature names are in the form {feature type}.{feature name}. Feature type abbreviations: rmd regional mutation density profiles, sigs mutational signatures, mut_load total number of single base substitutions, double base substitutions or indels, gene presence of gene gain or loss of function events, chrom_arm chromosome arm copy number fold change versus overall genome ploidy, genome genome properties including genome ploidy, diploid proportion, whole-genome duplication status, gender sample gender as determined by copy number data, sv structural variants, fusion presence of gene fusions, viral_ins presence of viral sequence insertions. Cancer type abbreviations, CNS central nervous system, CNS_Medullo medulloblastoma, CNS_PiloAstro pilocytic astrocytoma, NET neuroendocrine tumor, Sarcoma_GIST gastrointestinal stromal tumor, Sarcoma_Leiomyo leiomyosarcoma, Sarcoma_Lipo liposarcoma, Sarcoma_Osteo osteosarcoma, Sarcoma_Other sarcomas other than leiomyosarcoma, liposarcoma or gastrointestinal stromal tumors.

cancer types (middle panel). Also shown are the corresponding feature values in the patient in relation to the average feature value amongst patients in the training set (right panel).

We will use patient HMF004048A as an exemplary demonstration of the graphical report (Fig. 4a). Since the Lung_NonSmall-Cell probability (0.96) was much higher than the probability of other cancer types, only one cancer type (i.e. panel row) is shown. Up to three panel rows can be shown when more than one cancer type probability is high (such as for patient DO7304; Fig. 4b). One of the top features for HMF004048A was the presence of an *EML4-ALK* fusion (middle panel) which is on average found in ~4% of Lung_NonSmallCell patients (pink label), but only ~0.1% in all other patients (blue label). Since this feature is of boolean type, a feature value of 1 (red label) indicates the presence of the *EML4-ALK* fusion in HMF004048A (whereas a feature value of 0 would indicate absence). Additionally, the contribution of the non-small cell lung cancer RMD profiles (rmd.Lung_NonSmall-Cell.2) as well as the contribution of the APOBEC-associated

signatures SBS2 and SBS13[20] in HMF004048A (red labels) is higher than in Lung_NonSmallCell patients (pink labels), but also compared to all other patients (blue label). Whether a feature value in the patient of interest is higher or lower than that in all other patients is also indicated in the text in the feature contribution panel (middle panel) for non-boolean features.

Patient DO7304 (Fig. 4b) is an example of a situation where more than one cancer type probability is high, with the probability of Lymphoid being 0.78 and CNS_PiloAstro being 0.75 (Fig. 4b). Here, two feature panels are shown for both of these cancer types to aid with resolving this uncertainty. Since *IGLL5* loss is specific to lymphomas[48], we can confirm that the likely correct prediction is indeed Lymphoid.

This graphical report presents the detailed machine learning-based classification output of CUPLR in a human-readable format. We acknowledge that the output is highly detailed, which is inevitable due to the large amounts of data used by the algorithm. However, as these details may not be necessary for all

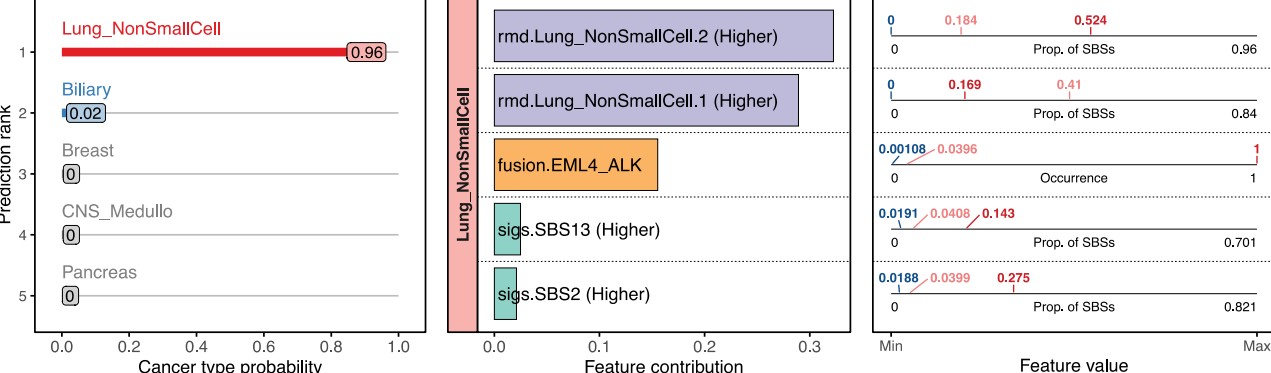

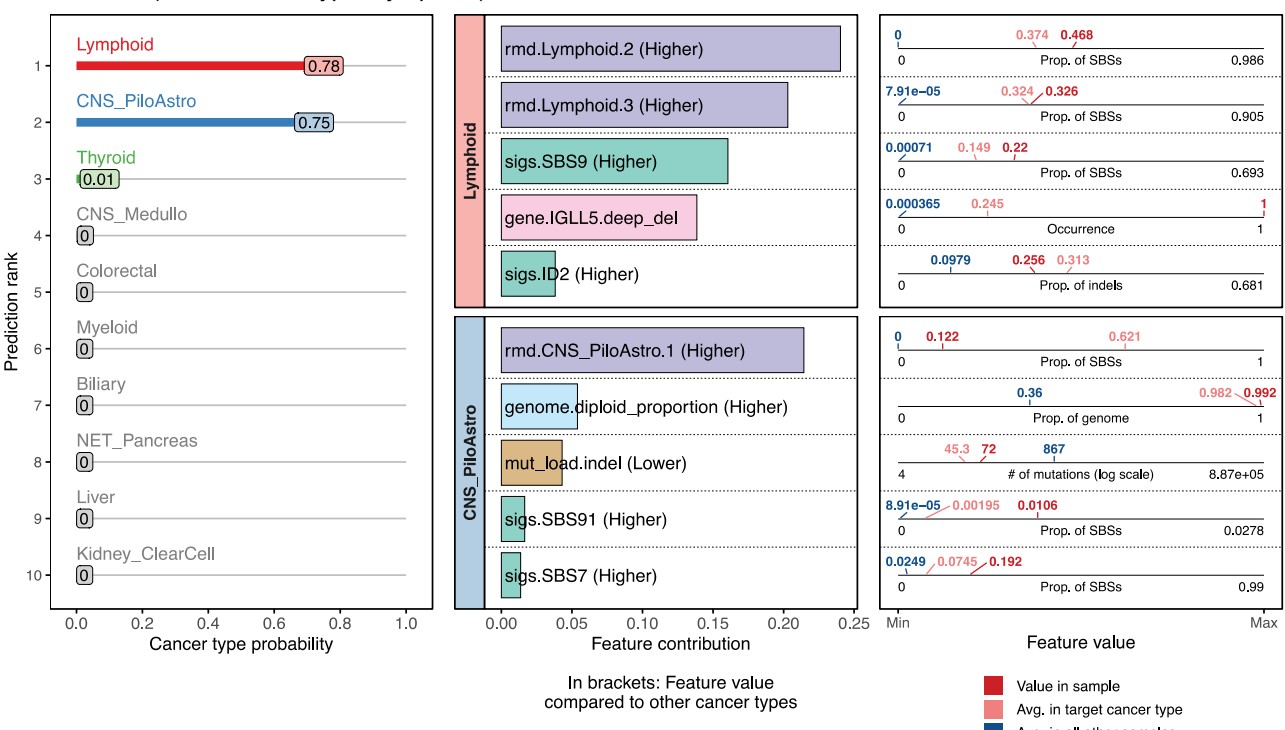

**Fig. 4 Graphical report for CUPLR predictions.** Example reports are shown for two patients from the holdout set: **a** HMF004048A, annotated as having non-small cell lung cancer (Lung_NonSmallCell), and **b** DO7304, annotated as having lymphoma (Lymphoid). The leftmost panels show the predicted cancer type probabilities. In the middle panels, contributions of the top features are shown for the top predicted cancer types. When there is uncertainty in the cancer type probabilities such as in **b**, more feature contribution panel rows are shown. In the rightmost panel, the feature values in the patient (red label) are plotted in relation to the average feature value amongst patients in the training set with the target cancer type (pink label) as well as all other samples not belonging to the target cancer type (blue label). For a full description of each feature, see Supplementary data 3. Feature type abbreviations: rmd regional mutation density profile, sigs mutational signatures, mut_load total number of single base substitutions, double base substitutions or indels, gene presence of gene gain or loss of function events, genome genome properties including genome ploidy, diploid proportion, whole-genome duplication status, fusion presence of gene fusions.

circumstances, we have implemented the option in the software to hide the feature contribution and/or feature value panels in the final graphical output.

**Feature contributions aid in clarifying the primary tumor location of CUPs.** To showcase how CUPLR could be used in a real-life clinical setting, we applied CUPLR to 141 tumors with a CUP diagnosis from the Hartwig dataset. From these, we could confidently ($n = 68$) or partially ($n = 14$) resolve the cancer type for 82 (58%) patients by examining the top predicted cancer types

and corresponding top contributing features for each patient (Supplementary data 5).

Of the 68 patients with fully resolved cancer types, 44 exhibited features with well-known cancer type associations in combination with high contribution of respective cancer type RMD profile. This included: four patients predicted as Breast with breast cancer-specific driver mutations (in *CHD1*, *GATA3*, *PIK3CA*, and/or *ZNF703*)[49]; 7 patients predicted as Colorectal with APC mutations[50] and/or the presence of the colibactin signature (SBS88)[51]; 5 patients predicted as Gastroesophageal with the ROS damage signature SBS17[20], LINE insertions[52], and/or *FHIT*

deletions[53]; 21 patients predicted as Lung_NonSmallCell with signatures of smoking (SBS4, DBS2 and/or ID3[20]); 3 patients predicted as Pancreas with *KRAS* mutations[50]; 1 patient predicted as Prostate with a *TMPRSS2-ERG* fusion[39]; 2 patients predicted as Urothelial with *TERT* mutations[54]; and 1 patient predicted as Uterus with a *PIK3R1* mutation[55]. The remaining 24 patients with a top cancer type probability ≥0.8 were likely correctly classified as most of these patients were predicted as high recall cancer types (>0.9, Fig. 2) with highly specific RMD profiles respective to each cancer type, including Breast Colorectum, Gastroesophageal, Kidney_ClearCell, Liver, Kidney_ClearCell, and Ovarian (Supplementary data 5, Supplementary fig. 11).

For the 14 patients with partially resolved cancer types, we could only determine the cancer supertype. For example, four patients had >0.6 probability of both Lung_NonSmallCell and Lung_SmallCell and exhibited smoking signatures SBS4, DBS2, and/or ID3[20], indicating that these patients most likely had lung cancer, though the subtype remains uncertain. Likewise, we could narrow down the TOO to 2 probable cancer types for 3 patients. For example, patient HMF002806A had >0.5 probability of Uterus and Breast, and had mutations in *PIK3CA* and *PTEN* which are common in both these cancer types[50]. This is indicative of gynecological cancer which often is treated in a similar manner[56,57].

We thus demonstrate that CUPLR can potentially clarify the TOO for over half of patients with CUP. It is important to note that even if the TOO is only partially resolved for a patient, such a patient could now potentially have more treatment options. However, for the CUP patients discussed above, additional evidence would be required for validation and final diagnosis, which could for example be based on histopathological examinations. These were unfortunately not available for the retrospectively analyzed samples included here, but such information would be readily available in routine diagnostics.

## Discussion

Here, we have developed a tissue of origin classifier (CUPLR) for the analysis and diagnostics of CUPs using a large uniformly processed WGS dataset of both primary and metastatic tumors. Our classifier incorporates genomic features derived from SVs, as well as provides human interpretable explanations alongside each prediction which allows for manual resolving of CUPs, especially in cases of lower-scoring samples or for samples for which multiple tumor types are suggested by CUPLR.

Current state-of-the-art WGS-based classifiers, including those by Jiao et al.[6] and Salvadores et al.[7], achieve high recall (≥80%) for over three-quarters of the cancer types they predict by primarily employing RMD and mutational signatures as features, which are derived from simple mutations. CUPLR builds upon these classifiers, with the inclusion of SV and driver gene-related features improving performance for certain cancer types, such as pilocytic astrocytoma and prostate cancer. Of note, genomic-based TOO classifiers published thus far have only used data from primary tumors[4–7]. However, since CUPs are by definition metastatic tumors, the inclusion of 4391 metastatic tumors with known tissue of origin for training CUPLR may make it a more suitable tool for the purpose of clarifying CUPs. We do acknowledge however that the metastatic tumor data used here may harbor treatment effects that are absent in treatment-naïve CUPs, such as driver mutations in *AR*[58] or *ESR1*[59] conferring resistance to therapy or the presence of characteristic mutations induced by for example 5-fluorouracil[36] or radiotherapy[37,60]. Identification and removal of such treatment-associated features could potentially improve TOO classification. Additionally, CUPLR is able to distinguish the largest number of cancer (sub)types (35 cancer

types) compared to existing WGS-based classifiers, with the Jiao[4] and Salvadores[5] classifiers discriminating 24 and 18 cancer types respectively, and also more than a recently published whole histology slide image-based classifier which discriminates 18 cancer types[61]. We do acknowledge that discriminating even more cancer types is warranted, but this is currently limited by the number of available training samples that were sequenced and uniformly bioinformatically analyzed. For example, thymic cancer had too few (<15) samples to be included as a separate class for training (Supplementary data 1). Furthermore, certain cancer types could be divided into their subtypes, such as Myeloid (currently only with 34 samples in total; Fig. 2a) into acute myeloid leukemia and multiple myeloma[62], and sarcomas in a broader range of subtypes[63]. The availability of more WGS data for less frequent, but also (ultra-)rare cancers would allow for the training of an updated CUPLR model that classifies additional cancer types and subtypes.

While CUPLR achieved overall excellent recall (90%) which is similar to or better than other WGS-based classifiers (Supplementary fig. 9), it should be noted that this is driven by several large sub-cohorts of common cancer types (e.g., breast and colorectal cancer) and that performance for certain cancer types is still suboptimal. For cancer subtypes that CUPLR has difficulty discriminating, such as small cell versus non-small cell lung cancer and papillary versus clear cell renal carcinoma, additional information from histopathological stainings could be used to clarify these cases. Here, the application of artificial intelligence-based histology image analysis[61] could further improve the prediction performance and reliability of resolving CUPs. Clinical metadata, such as biopsy location and metastasis grade[64], when used together with CUPLR can also aid in clarifying primary tumor location. For instance, there may be uncertainty as to whether a tumor with human papillomavirus DNA integration was a head and neck cancer or cervical cancer based on CUPLR predictions (e.g., DO51592 with a probability of Cervix = 0.754 and HeadAndNeck_Other = 0.310; Supplementary data 5), since human papillomavirus DNA integration is characteristic of these two cancer types[43]. However, in clinical practice, it will always be known if the tumor biopsy was taken from the upper body and if the lesions were local, and with this information, one can conclude that this tumor can only be of head and neck origin.

Given that RMD[6,7], mutational signatures[65], and SVs[27,66] are still active areas of research, we expect improvements to these features could also lead to better TOO classification. Here, we have demonstrated that extraction of cancer type-specific RMD profiles is possible from raw mutation counts, similar to what was done for mutational signatures[20]. However, CUPLR does not heavily rely on RMD profiles for the classification of certain cancer types such as liposarcoma and non-melanoma skin cancer (Fig. 3b), potentially because more training samples are required to extract more stable and informative RMD profiles which could improve classification for these cancer types. Improvements to TOO classification may also come from the extraction of more comprehensive mutational signatures, for example by incorporating information on mutation timing or genome localization[67,68]. Development of more sophisticated signature extraction methods may also allow for quantification of low signal tissue type-specific signatures, such as SBS88 (associated with colibactin-induced DNA damage in the colon) which has only been extracted from colon healthy tissue[51,69] but not cancer tissue likely because other mutational processes in cancer tissue mask the presence of this signature (Supplementary figs. 7 and 11). Lastly, while CUPLR uses a wide range of features derived from SVs (including gene fusions, viral DNA integrations, LINE insertions, structural deletion and duplication size, and chromothripsis), there is still an opportunity to explore other SV-related features to improve TOO classification, such as SV signatures[66].

Given that WGS is rapidly maturing and is now slowly being adopted in routine diagnostics for comprehensive molecular diagnostics[15,16], CUPLR serves as a viable complementary tool to standard procedures for determining TOO (e.g., histopathological stainings). CUPLR can be run from the output of open source tools and is freely available as an R package [https://github.com/UMCUGenetics/CUPLR]. The trained model as well as the code for generating the input features are provided to enable prediction on new samples and for facilitating diagnostic use.

## Methods

**Datasets**. We have matched tumor/normal whole-genome sequencing data from cancer patients from two cohorts: the Hartwig Medical Foundation (Hartwig) cohort and the Pan-Cancer Analysis of Whole Genomes (PCAWG) cohort.

The Hartwig cohort included 4902 metastatic tumor samples from 4572 patients. The data was provided under data request DR-104 from the Hartwig Medical Foundation. All patients in this resource have given consent for the reuse of their genomic and clinical data for research purposes. For patients with multiple biopsies that were taken at different timepoints, patient IDs were suffixed by 'A' for the first biopsy, 'B' for the second biopsy, etc (e.g., HMF001423A, HMF001423B). Normal samples (blood) had a mean read coverage of ~30× while tumor samples had coverage of ~90×[70]. Only a single sample of each patient was used for this study. To do this, we selected the tumor sample with the earliest biopsy date, and if this information did not exist we selected the sample with the highest tumor purity. However, some Hartwig patients had biopsies from different primary tumor locations. In these cases, we kept at least one sample from each primary tumor location, and when there were multiple samples from the same primary tumor location, we applied the aforementioned biopsy date and tumor purity filtering criteria.

The PCAWG cohort consisted of 2835 patient tumors. Access to raw sequencing data for these tumors was granted via the Data Access Compliance Office (DACO) Application Number DACO-1050905 on 6 October 2017 and via the Cancer Genome Collaboratory download portal [https://console.cancercollaboratory.org] on 4 December 2017. Normal samples (blood, adjacent tumors, or distant tumors) had a mean read coverage of 39×, while tumor samples had a bimodal coverage distribution with modes at 38× and 60×[50]. Samples with <0.2 tumor purity were excluded from this study as somatic variant calling was not reliable for these samples. PCAWG samples that were gray- or blacklisted by the PCAWG consortium were also excluded [https://dcc.icgc.org/releases/PCAWG/donors_and_biospecimens].

For both cohorts, we only kept samples with ≥50 SNVs/indels, and removed an additional set of samples for several reasons including failed variant calling, insufficient informed consent for use of the WGS data, and one duplicate PCAWG sample (DO217844) that was already included in the Hartwig cohort. Lastly, we only selected samples from cancer types with at least 15 samples. Ultimately, we selected 4391 Hartwig samples and 2365 PCAWG samples for training, as well as 141 Hartwig CUP samples for the CUP analysis (Supplementary data 1).

**Variant calling**. Somatic mutation data of the CPCT, DRUP, and WIDE projects were kindly shared by Hartwig on 6 February 2020 with an update received on 20 October 2021. To exclude technical noise from PCAWG and Hartwig somatic variant calling workflows, we have reanalyzed the PCAWG samples with the Hartwig pipeline for somatic variant calling [https://github.com/hartwigmedical/pipeline5] which was hosted on the Google Cloud Platform using Platinum [https://github.com/hartwigmedical/platinum][23]. Details of the full pipeline are described by Priestley et al.[70] as well as in the Hartwig pipeline GitHub page. Briefly, reads were mapped to GRCh37 using BWA (v0.7.17). GATK (v3.8.0) Haplotype Caller was used for calling germline variants in the reference sample. SAGE (v2.2) was used to call somatic single and multi-nucleotide variants as well as indels. GRIDSS (v2.9.3) was used to call SVs. PURPLE combines B-allele frequency (BAF) from AMBER (v3.3), read depth ratios from COBALT (v1.7), and structural variants from GRIDSS to estimate copy number profiles, variant allele frequency (VAF), and variant clonality. Additionally, PURPLE also determines the gender (based on sex chromosome ploidy), the proportion of the genome that is diploid, as well as the presence of whole-genome duplication in a sample. LINX interprets SVs (to identify simple and complex structural events) from PURPLE, and also detects gene fusions, viral DNA integrations, and homozygously disrupted genes. Importantly, we ensured that mutation (simple and complex) filtering and annotation tools were run with the same versioning for PCAWG and HMF cohort. For PURPLE we relied on v2.53 whereas for LINX we used v1.17.

### Extraction of features

*Regional mutational density*. RMD was defined as the number of somatic SBSs in each 1 Mb bin across the genome ($n = 3071$), normalized by the total number of SBSs in the sample. Extraction of RMD profiles from these RMD bins was performed within the CUPLR training procedure (Supplementary fig. 1a) using non-negative matrix factorization (NMF). This is described in detail in the "CUPLR

training procedure" methods section. See Supplementary data 6 for a visualization of each RMD profile.

*Mutational signatures*. The number of somatic mutations falling into the 96 SBS, 78 DBS and, 83 indel contexts (as described in COSMIC: [https://cancer.sanger.ac.uk/signatures/]) was determined using the R package mutSigExtractor [https://github.com/UMCUGenetics/mutSigExtractor], v1.23). To obtain the mutational signature contributions for each sample, the mutation context counts were fitted to the COSMIC catalog of mutational signatures using the nnlm() function from the NNLM R package.

The contributions of the child signatures SBS7a, SBS7b, SBS7c, and SBS7d were summed to yield the parent signature SBS7. Similarly, SBS10a-d and SBS17a-b were merged to yield SBS10 and SBS17. Lastly, the SBS, DBS, and indel signature contributions were normalized by the total number of SBSs, DBSs, and indels respectively.

*Chromosome arm ploidy*. Chromosome arm ploidies were determined in a similar method as described by Taylor et al. 2018[26].

Somatic copy number (CN) segments (called by PURPLE) were split by their respective chromosome arms. Only chr1-22 and chrX were included. All chromosomes have p and q arms, except for chr13, chr14, chr15, chr21, and chr22 which are considered to only have the $q$ arm. For each chromosome arm, the CN values of each segment were converted to integer values. The arm coverage of each CN integer value was then determined (e.g., 70% of the arm has a CN of 2, 20% a CN of 1, and 10% a CN of 3). The CN with the highest coverage was assigned as the preliminary arm CN. The most common CN across all arms was assigned as the genome CN.

Two filtering steps were then performed to obtain the final arm CN values. For each arm, if the CN with the highest coverage has <50% coverage, and if any of the CN values equal the genome CN, then assign that CN as the final CN of the arm. Else, assign the genome CN as the final CN of the arm.

To determine the CN gains and losses of each arm, the fold change between the arm CN and genome CN was calculated.

*Features extracted from LINX output*. LINX combines simple mutations (point mutations and indels), structural variants, and copy number variants to resolve simple and complex structural rearrangements, and subsequently identify gene driver events, and gene fusions and detect viral DNA integrations.

The presence of 4 types of gene driver events was determined from the output of LINX: (i) amplifications, (ii) deep deletions, (iii) biallelic loss, and (iv) monoallelic hits (pathogenic mutation in one allele). Amplified genes were marked as likelihoodMethod = ='AMP' by LINX. Genes with deep deletions were marked as likelihoodMethod = ='DEL'. Genes with biallelic loss were marked as biallelic = =TRUE. Genes with a monoallelic pathogenic mutation were marked as biallelic = =FALSE and driver = ="MUTATION", and we selected only those with driverLikelihood ≥ 0.9 (referring to the likelihood of the mutation being impactful as determined by the dndscv R package[71]). LINX determines the presence of driver events for 462 genes. Thus, there are 462 genes × 4 driver types = 1848 gene driver features in total. We then performed preliminary feature selection to reduce the computational resources required for training CUPLR. Here, one-sided Fisher's exact tests were performed and Cramer's V values were calculated. Only genes where at least one driver type had a $p$ value <0.01 and Cramer's V ≥ 0.1 were kept (203 genes × 4 driver type = 812 gene driver features).

Gene fusions belonged to 3 categories: (i) well-known fusion pairs (e.g., *TMPRSS2-ERG*), (ii) immunoglobulin heavy chain (IGH) locus fusions, and (iii) fusions with promiscuous gene partners (e.g., *BCR*). *IGH* fusions were grouped into a single feature as these are characteristic events in lymphoid cancers[72]. Fusions with one promiscuous gene partner were grouped by gene (e.g., *RUNX1_ETS2* and *RUNX1_RCAN1* would both fall under the *RUNX1_\** feature). Fusions with two promiscuous gene partners were split into two separate features (e.g., *SLC45A3_MYC* would become the features *SLC45A3_\** and *\*_MYC*). Only fusions that were marked as reported = =TRUE by LINX (i.e., reported in the literature) were selected. We then performed preliminary feature selection due to a large number of possible fusions present in our dataset ($n = 512$). Here, one-sided Fisher's exact tests were performed and Cramer's V values were calculated. Only fusions with $p$ value <0.01 and Cramer's V ≥ 0.1 were kept (46 fusions).

For the viral DNA integrations present in our dataset, we merged virus strains into nine virus categories: adeno-associated virus (AAV), Epstein-Barr virus (EBV), hepatitis B virus (HBV), hepatitis C virus (HCV), human immunodeficiency virus (HIV), human papillomavirus (HPV), herpes simplex virus (HSV), human T-lymphotropic virus (HTLV), and Merkel cell polyomavirus (MCPyV). For example, *human papillomavirus type 16* and *human papillomavirus type 18* would be both grouped as *human papillomavirus*.

LINX chains individual SVs into SV clusters and classifies these clusters into various event types. Clusters can have one SV (for simple events such as deletions and duplications), or multiple SVs. We defined SV load as the total number of SV clusters. We quantified the presence of several SV types including: (i) deletions and duplications (ResolvedType is 'DEL' or 'DUP') stratified by length (1–10 kb, 10–100 kb, 100kb–1 Mb, 1–10 Mb, >10 Mb), (ii) complex SV clusters (ResolvedType = ='COMPLEX') stratified by the number of clusters (0–25, 25–50,

50–100, 100–200, 200–400, 400–800, >800), (iii) long interspersed nuclear element (LINE) insertions (ResolvedType = ='LINE'), and double minutes (ResolvedType = ='DOUBLE_MINUTE'). Lastly, we also determined the number of breakends for the complex cluster with the most breakends.

## CUPLR training procedure

*Extraction of regional mutational density profiles.* To extract the cancer-specific RMD profiles from the 3071 RMD bins, a multistep procedure involving non-negative matrix factorization (NMF) (Supplementary fig. 2) was performed prior to classifier training (Supplementary fig. 1ai).

All NMF runs described below are performed with the nnmf() function from the NNLM R package (v0.4.4) with the parameters loss = 'mkl' and max.iter=2000.

For each cancer type cohort, an NMF rank search was done to determine the optimum rank (i.e. number of RMD profiles) (Supplementary fig. 2a). For ranks 1 to 10, NMF was performed 50 times on a random subset of 100 samples from the cohort (or if the cohort contained less than 100 samples, all samples from that cohort were used) with 10% of the values randomly removed. The missing values were then imputed and the mean squared error (MSE) of these imputed values was calculated. This method of calculating MSE is described by the authors of the NNLM R package[73]. The median of the MSE across the 50 NMF iterations was then calculated. The rank search thus results in 10 MSE values across the 10 ranks searched. The optimum rank was the rank before the increase in log10(MSE) was >0.2%, and NMF was then performed using the optimum rank and without removing random values to produce the RMD profiles for the cancer type cohort (Supplementary fig. 2b).

The above procedure thus yields a different set of RMD profiles for each cancer type cohort. However, some RMD profiles across related cancer types (e.g., pancreas and biliary cancer) may actually be equivalent RMD profiles. Hierarchical clustering (using Pearson correlation as a distance measure) was thus performed to group similar RMD profiles across all cancer-type cohorts. The resulting dendrogram was cut at a height of 0.1 (using the R function cutree()), whereby RMD profiles under this height were grouped and considered the same profile. From each of the groups, one profile was greedily selected to yield the final set of RMD profiles (Supplementary fig. 2c).

To obtain the RMD profile contributions for each sample, the RMD bins were fitted to the RMD profiles using the nnlm() function from the NNLM R package.

*Random forest ensemble training.* The main component of CUPLR comprises an ensemble of binary random forests that each discriminates one cancer type (Supplementary fig. 1aii). The below text describes the training procedure for each cancer type random forest.

First, univariate feature selection was performed to remove irrelevant features (Supplementary fig. 1aiii). Pairwise testing was done to compare feature values from samples of the target cancer type (case group) versus the remaining samples (control group). For numeric features, *p* values were determined using Wilcoxon rank-sum tests, and effect sizes were calculated using Cliff's delta. For boolean features, *p* values were determined using Fisher's exact tests, and effect sizes were calculated using Cramer's V. Depending on the feature, alternative hypotheses for the Wilcoxon rank-sum tests and Fisher's exact tests were one or two-sided. See Supplementary Data 3 for details on which features are numeric or boolean, as well as which alternative hypothesis was used. Features were kept which had $p < 0.01$ and effect size ≥0.1. The number of features kept was capped to 100 features.

Second, random oversampling was performed for the case group which always contained fewer samples than the control group, which was randomly undersampled (Supplementary fig. 1aiv). A grid search was performed to determine the optimal pair of 5 oversampling and 5 undersampling ratios. These ratios were automatically determined as follows: i) calculate the geometric mean between the case and control group sample sizes; ii) the resampling ratios are logarithmically spaced between the geometric mean and the case group sample size or the control group sample size. For each over-/undersampling ratio pair, stratified 10-fold cross-validation (CV) was performed, after which the area under the precision-recall curve (AUPRC) was calculated. The pair with the highest AUPRC was chosen and the resampling was applied. CV and AUPRC calculations were performed using the mltoolkit R package [https://github.com/UMCUGenetics/mltoolkit].

Lastly, a random forest was trained (Supplementary fig. 1av) using the randomForest R package (v4.6-14) with default settings. A filter is applied to the probabilities produced by the random forest based on sample gender, where breast, ovary and cervix probabilities are set to 0 for male samples, and prostate probabilities are set to 0 for female samples. Local increments were calculated for the random forest using the rfFC R package (v1.0) to enable downstream calculation of feature contributions[47].

*Isotonic regression training.* The entire random forest ensemble training procedure was then subjected to stratified 15-fold cross-validation which allows every sample to be excluded from the training set in order to obtain cancer type probabilities for the training samples (Supplementary fig. 1b). These cross-validation probabilities were then used to train an ensemble of isotonic regressions using the isoreg() R function (one per cancer type random forest) to calibrate the probabilities produced by the random forest ensemble (Supplementary figs. 1c and 3).

Random forests tend to be overconfident at probabilities towards 0 and underconfident at probabilities towards 1[28], and this bias varies between random forests (Supplementary fig. 4). In other words, a probability of e.g., 0.8 from one random forest does not correspond to a probability of 0.8 from another random forest. Probability calibration greatly reduced this bias ensuring that predictions across the random forests are comparable (Supplementary fig. 4).

*Performance evaluation.* To assess the performance of CUPLR, we used the cancer type predictions based on the isotonic regression calibrated cross-validation probabilities, as well as by applying the final model to a validation set whereby 10% of samples were held out from the full training set (Supplementary fig. 1). Performance metrics per cancer type were defined as follows (using 'Breast' as an example):

Recall = Fraction of Breast samples predicted as Breast
Top-2 recall = Fraction of Breast samples where the 1st or 2nd top prediction was Breast
Precision = Amongst samples predicted as Breast, the fraction of samples predicted as Breast

Overall performance metrics were micro-averages of the per-cancer-type metrics and defined as follows:

Micro-averaged recall = Accuracy = Fraction of all samples correctly predicted
Micro-averaged top-2 recall = Top-2 accuracy = Fraction of all samples where the 1st or 2nd highest probability prediction was correct
Micro-averaged precision = Micro-average of per-cancer-type precision

Precision and recall curves for each binary random forest classifier within CUPLR are shown in Supplementary fig. 12.

We used predictions based on calibrated cross-validation probabilities to assess the effect of excluding feature types on recall (Supplementary fig. 8); compare the recall of CUPLR to other classifiers (Supplementary fig. 9); assess the effect of confounding factors on recall (Supplementary table 1, Supplementary fig. 13, Supplementary Notes); as well as to show that 30× coverage is sufficient for reliable CUPLR predictions for most cancer types, with ≥ 60× coverage giving the most reliable predictions (Supplementary fig. 14, Supplementary Note 2).

Lastly, we showed that there was likely minimal overfitting on the Hartwig and/ or PCAWG data sets (i.e., batch effects) by training a CUPLR-like model solely on Hartwig samples and another model solely on PCAWG samples, and thereafter determining performance by testing on the opposite data set (Supplementary fig. 15, Supplementary Note 3).

**Reporting summary.** Further information on research design is available in the Nature Research Reporting Summary linked to this article.

## Data availability

For the Hartwig cohort, WGS data and corresponding metadata have been obtained from the Hartwig Medical Foundation and provided under data request number DR-104. Both WGS data and metadata are freely available for academic use from the Hartwig Medical Foundation through standardized procedures and request forms which can be found at [https://www.hartwigmedicalfoundation.nl]. Somatic variant calls, gene driver lists, copy number profiles, and other core data of the PCAWG cohort generated by the Hartwig analytical pipeline are available for download at [https://dcc.icgc.org/releases/PCAWG/Hartwig]. Researchers will need to apply to the ICGC data access compliance office [https://daco.icgc-argo.org] for the ICGC portion of the dataset. Authentication of NIH eRA commons is required to access the TCGA portion of the dataset via [https://icgc.bionimbus.org]. Additional information on accessing the data, including raw read files, can be found at [https://docs.icgc.org/pcawg/data/]. Metadata for PCAWG samples (e.g., sample whitelisting) can be found at https://dcc.icgc.org/releases/PCAWG. The extracted features for each sample and used to develop CUPLR are available at https://doi.org/10.5281/zenodo.5939805[74]. All other processed and raw data can be found in the Supplementary Data files.

## Code availability

The Hartwig Medical Foundation pipeline [https://github.com/hartwigmedical/pipeline5], hosted on the Google Cloud Platform using Platinum [https://github.com/hartwigmedical/platinum], was used for germline and somatic variant calling, as well as post-processing procedures such as identification of simple and complex structural rearrangements, annotation of driver gene mutation events, and detection of gene fusions and viral DNA integrations. CUPLR can be run from the output of this pipeline, and is available as an R package on GitHub ([https://github.com/UMCUGenetics/CUPLR]; https://doi.org/10.5281/zenodo.6637693[75]). This repository also contains the code for data processing and generating the figures in this paper. CUPLR depends on some custom code, including mutSigExtractor for extraction of mutational signatures [https://github.com/UMCUGenetics/mutSigExtractor] and mltoolkit (only required for classifier training and not for running CUPLR; https://github.com/UMCUGenetics/mltoolkit).

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

## Acknowledgements

This research was supported by a grant (2020-Cuppen) of Stichting Hanarth Fonds, The Netherlands ([www.hanarthfonds.nl]; received by E.C.). This publication and the underlying study have been made possible partly on the basis of the data that Hartwig Medical Foundation and the Center of Personalized Cancer Treatment (CPCT) have made available to the study.

## Author contributions

L.N. performed analyses, wrote/edited the paper. A.V.H. conceived the study, performed analyses, wrote/edited the paper. E.C. edited the paper and provided discussion. E.C. and A.V.H. supervised the study. All authors proofread, made comments, and approved the paper.

## Competing interests

The authors declare no competing interests.
