## [Peer Review File · Nature Communications]

This manuscript has been previously reviewed at another journal that is not operating a transparent peer review scheme. This document only contains reviewer comments and rebuttal letters for versions considered at *Nature Communications*.

REVIEWER COMMENTS

Reviewer #1 (Remarks to the Author):

The authors modified the manuscript according to my previous suggestions.

I have only a minor concern about reference 12, which does not cover the mentioned topic (CUP TOO prediction using microRNA signatures). Please select proper references (see PMID: 22618571 PMID: 34075699).

Reviewer #4, commented on your responses to Reviewer's #3 previous concerns (Remarks to the Author):

This study presents a classification tool for tumor site of origin using whole-genome sequencing data using off-the-shelf machine learning algorithms. The analysis differs from existing work in the inclusion of structural variants. Some of the findings on the SV side include the chromothripsis in lipo-sarcoma, TMPRSS2_ERG for prostate cancer. Many of the SV events discussed are well known and can be detected using whole-exome and targeted panels. It is unclear what's new from this whole-genome analysis. Overall, I appreciate the amount of work presented in this study but not convinced that this study presented sufficient technical or scientific advance over existing work.

A major technical limitation is that there is substantial imbalance in sample size in this dataset as reviewer 3 pointed out, and many cancer types and subtypes have insufficient sample sizes to conduct proper hold-out test and even cross-validation. The overall accuracy measure can be misleadingly high in cases with high imbalance of case and "control". The analysis should restrict to cancer types that have sufficient sample size. In addition, sensitivity and PPV should be reported as a more honest assessment of the classification performance. For rare cancer types a classifier should take into account of the prevalence of the disease.

Despite the combined PCAWG and Hartwig cohort, the sample size is still far from sufficient to establish a classifier for all 30+ tumor types. The clinical utility is also unclear due to the limited availability of WGS in clinical setting.

RESPONSE TO REVIEWER COMMENTS

Reviewer #1 (Remarks to the Author):

The authors modified the manuscript according to my previous suggestions.

I have only a minor concern about reference 12, which does not cover the mentioned topic (CUP TOO prediction using microRNA signatures). Please select proper references (see PMID: 22618571 PMID: 34075699).

We thank the reviewer for the reference recommendations and have added these to the text.

Reviewer #4, commented on your responses to Reviewer's #3 previous concerns (Remarks to the Author):

This study presents a classification tool for tumor site of origin using whole-genome sequencing data using off-the-shelf machine learning algorithms. The analysis differs from existing work in the inclusion of structural variants. Some of the findings on the SV side include the chromothripsis in lipo-sarcoma, Tmprss2_erg for prostate cancer. Many of the SV events discussed are well known and can be detected using whole-exome and targeted panels. It is unclear what's new from this whole-genome analysis. Overall, I appreciate the amount of work presented in this study but not convinced that this study presented sufficient technical or scientific advance over existing work.

We respectfully disagree with the view of the reviewer. While the tumor-specific biological observations underlying the classifier are indeed not novel, this is also not the scope of the paper. We here describe a novel tissue of origin classifier that can be used in a diagnostic setting. This classifier uses previously described principles, but also includes novel features. Although such features have been described in other contexts, they have never been shown to be informative in a pan-cancer tissue of origin classification approach.

We agree that some of the SV events such as fusions are detectable with other DNA or RNA-based sequencing approaches. However, some SV events, such as LINE insertions (important for classifying gastric and urothelial cancers) or viral sequence insertions (important for classifying skin carcinoma, cervical cancer and head/neck cancer) are not detectable (or routinely detected) by whole-exome/targeted sequencing. Furthermore, the regional mutation density (RMD) features do require WGS data and we demonstrate that these are the most predictive features for tumor type classification overall.

The strength of WGS is that only one sequencing assay is needed to extract both RMD and SV features (but also other genomic features). For example, while it is true that Tmprss2_erg fusions as well as AR amplifications are (potentially) detectable with whole-exome/targeted sequencing and are defining characteristics of prostate cancer, not all prostate cancers carry these events. In our cohort, 209/560 (37%, see below table) of prostate patients have neither of these events.

Cancer type classification for these 209 patients could still be performed based on the prostate RMD profile which is highly predictive of prostate cancer (rmd.Prostate.1, see barplot below). This data is readily available with WGS, whereas with whole-exome/targeted sequencing additional assays would need to be performed to come to the same conclusion for each individual patient tested.

TMPRSS2_ERG	AR_amp	Freq
FALSE	FALSE	209
TRUE	FALSE	217
FALSE	TRUE	70
TRUE	TRUE	64

Frequency of TMPRSS2-ERG fusions and/or AR amplifications in prostate cancer samples

Excerpt from Supplementary figure 7: Feature importances for the 'Prostate' random forest within CUPLR.

A major technical limitation is that there is substantial imbalance in sample size in this dataset as reviewer 3 pointed out, and many cancer types and subtypes have insufficient sample sizes to conduct proper hold-out test and even cross-validation. The overall accuracy measure can be misleadingly high in cases with high imbalance of case and "control". The analysis should restrict to cancer types that have sufficient sample size. In addition, sensitivity and PPV should be reported as a more honest assessment of the classification performance. For rare cancer types a classifier should take into account of the prevalence of the disease.

Regarding accuracy and class imbalance:

It is indeed correct that accuracy would be misleading if we would have calculated *total* accuracy: (number of samples correctly predicted as e.g. breast) / (number of samples across all cancer types). However, we have calculated *per cancer type* accuracy which is not affected by class imbalances: (number of samples correctly predicted as breast) / (total number of breast samples). However, to better illustrate the performance per tumor type, we have now added sensitivity and PPV curves for each cancer type binary classifier in CUPLR as Supplementary figure 11 to the revised manuscript.

Regarding sample sizes:

In our training set, 32/35 cancer types had ≥ 30 samples, with the smallest cancer type cohort (NET_Lung) having 23 samples (see table below). We believe that this is sufficient sample size to assess CV performance. We do agree that there is insufficient sample size to properly

calculate performance for certain cancer types on the independent test set. However, we observe that the CV and test set accuracies across cancer types are generally concordant (see Figure 2d below), suggesting that CV accuracy adequately represents test set accuracy. For these reasons, we have chosen to report both CV and test set accuracy in Figure 2d so that one can rely on CV accuracy for cancer types with low test set sample size.

cancer_type	n_test	n_train
Breast	100	896
Colorectal	71	639
Lung_NonSmallCell	58	530
Prostate	56	504
Skin_Melanoma	40	366
Pancreas	35	312
Liver	34	306
Gastroesophageal	33	295
Ovarian	28	256
Kidney_ClearCell	24	213
Lymphoid	21	197
Urothelial	20	180
CNS_Medullo	15	123
CNS_Glioma	13	118
Sarcoma_Other	13	118
NET_Pancreas	12	107
Biliary	12	102
HeadAndNeck_Other	9	82
Uterus	7	68
Thyroid	6	61
Sarcoma_GIST	7	60
Sarcoma_Leiomyo	7	57
Cervix	6	53
CNS_PiloAstro	6	52
NET_Gastrointestinal	5	49
Lung_SmallCell	5	48
Sarcoma_Osteo	4	40
Kidney_Chromophobe	4	39
Sarcoma_Lipo	4	38
Mesothelium	4	37
HeadAndNeck_SalivaryGland	3	31
Myeloid	4	30
Kidney_Papillary	3	29
NET_Lung	3	23
Skin_Carcinoma	2	23

Training and test set sample sizes for each cancer type, sorted by training sample size.

Figure 2: a) Training and test set sample sizes for each cancer type. b,c) Top-1 and top-2 accuracy as well as d) the confusion matrix based on cross-validation (left bars/cells) and test set (right bars/cells) predictions.

Despite the combined PCAWG and Hartwig cohort, the sample size is still far from sufficient to establish a classifier for all 30+ tumor types.

We acknowledge that increased sample size would of course be more ideal for classifier training for some cancer types, such as Kidney_Papillary, NET_Lung, Skin_Carcinoma, HeadAndNeck_SalivaryGland, Myeloid (which all have ≤ 31 training samples and $\leq 70\%$ accuracy based on either CV or test set; Figure 2a and d).

Low sample sizes are mainly the result of splitting cancer types into subtypes, which is often clinically relevant but ignored in many previously described classifiers. It should be noted that for many cancer types, CUPLR still predicts the correct supertype. For example, Kidney_Papillary is most often misclassified as Kidney_ClearCell, meaning that the cancer supertype (kidney) is still correctly predicted (Figure 2d). Likewise, NET_Lung is most often misclassified as other neuroendocrine tumors (NET_Gastrointestinal or NET_Pancreas). Additionally, other cancer types with relative few but slightly more training samples (< 60 samples) also have frequently correct supertype (but not subtype) predictions, such as Lung_SmallCell (misclassified as Lung_NonSmallCell), or Sarcoma_Osteo/Sarcoma_Leiomyo (misclassified as other sarcomas). However, in most cases, CUPLR more often predicted the subtype correctly than incorrectly and we thus believe that CUPLR can still provide highly relevant information to support diagnosis despite low sample sizes for certain cancer (sub)types.

From a clinical perspective, when dealing with CUPs, some indication of the potential cancer type (e.g. from CUPLR) is already better than no indication (which would often have been the original and final diagnosis). Additionally, it is important to note that CUPLR is intended to complement other clinical evidence (e.g. from histopathological stainings or patient metadata) to determine tissue of origin, rather than to come to the final decision of the tissue of origin by itself. Since CUPLR outputs explanations as to which features support a prediction, existing knowledge can be used to reject potentially bad predictions on a case by case basis. This reasoning is included in the discussion.

The clinical utility is also unclear due to the limited availability of WGS in clinical setting. As in the response to reviewer #1, we acknowledge that WGS is not yet standard in routine diagnostics in all hospitals in all countries and therefore CUPLR is not (yet) applicable for every situation. However, we see increasingly more labs and countries, especially in Europe, developing WGS-based cancer diagnostics programs [Campbell et al 2019] and the tool described here is targeting those developments. Of note, in the Netherlands, WGS (and not panel-based sequencing) is a reimbursed diagnostic approach for CUP patients since 2021 and used by an increasing number of hospitals despite the fresh-frozen material requirement, clearly illustrating that the current real-world scenario is changing and future directions likely include WGS.

REVIEWER COMMENTS

Reviewer #4 (Remarks to the Author):

There is some fundamental confusion on the definition of classification accuracy used in the study.

The authors defined a per sample accuracy as (number of samples correctly predicted as breast) / (total number of breast samples). This is the definition of "sensitivity" or "recall", NOT "overall accuracy".

The missing side of the equation is "precision" or "PPV" which is the fraction of actual breast cancer cases out of predicted breast cancer of origin.

Both precisions and recalls should be reported in the main text and a direct comparison of these values to those reported in the published whole-genome TSO studies they cited in the paper (ref 6 and 7) would more convincingly show the added value of this study.

RESPONSE TO REVIEWER COMMENTS

Reviewer #4

There is some fundamental confusion on the definition of classification accuracy used in the study.

The authors defined a per sample accuracy as (number of samples correctly predicted as breast) / (total number of breast samples). This is the definition of “sensitivity” or “recall”, NOT “overall accuracy”.

To avoid any further confusion, we have replaced (per-class) “accuracy” with “recall” in the manuscript. In the methods (under “Performance evaluation”) we have added the following text to define our performance metrics:

Performance metrics per cancer type were defined as follows (using ‘Breast’ as an example):

- *Recall = Fraction of Breast samples predicted as Breast*
- *Top-2 recall = Fraction of Breast samples where the 1st or 2nd top prediction was Breast*
- *Precision = Amongst samples predicted as Breast, fraction of samples predicted as Breast*

Overall performance metrics were micro-averages of the per-cancer-type metrics and defined as follows:

- *Micro-averaged recall = Accuracy = Fraction of all samples correctly predicted*
- *Micro-averaged top-2 recall = Top-2 accuracy = Fraction of all samples where the 1st or 2nd highest probability prediction was correct*

The missing side of the equation is “precision” or “PPV” which is the fraction of actual breast cancer cases out of predicted breast cancer of origin.

- *Micro-averaged precision = Micro-average of per-cancer-type precision*

Both precisions and recalls should be reported in the main text and a direct comparison of these values to those reported in the published whole-genome TOO studies they cited in the paper (ref 6 and 7) would more convincingly show the added value of this study.

We have included precision numbers in Figure 2 (see below) and mentioned these in the manuscript.

We have also included a comparison of CUPLR's precision to the classifiers in ref 6 (PCAWG neural network) and ref 7 (SVM from Salvadores et al) as Supplementary figure 10 (see below). We have mentioned in the manuscript that precision was overall similar to or better than the other classifiers, except for head and neck, myeloid, thyroid, and uterine cancers.